# Quantitative Volumetric Analysis Using 3D Ultrasound Tomography for Breast Mass Characterization

**DOI:** 10.3390/tomography11100111

**Published:** 2025-09-30

**Authors:** Maria L. Anzola, David Alberico, Joyce Yip, James Wiskin, Bilal Malik, Raluca Dinu, Belinda Curpen, Michael L. Oelze, Gregory J. Czarnota

**Affiliations:** 1Department of Radiation Oncology, Sunnybrook Health Sciences Centre, Toronto, ON M4N 3M5, Canada; marialourdes.anzolapena@sunnybrook.ca (M.L.A.); david.alberico@sunnybrook.ca (D.A.); waiszejoyce.yip@sunnybrook.ca (J.Y.); 2Physical Sciences, Sunnybrook Research Institute, Toronto, ON M4N 3M5, Canada; 3QT Imaging Holdings, Inc., Novato, CA 94949, USA; james.wiskin@qtimaging.com (J.W.); bilal.malik@qtimaging.com (B.M.); raluca.dinu@qtimaging.com (R.D.); 4Medical Imaging, Sunnybrook Health Sciences Centre, Toronto, ON M4N 3M5, Canada; belinda.curpen@sunnybrook.ca; 5Grainer College of Engineering, University of Illinois at Urbana-Champaign, Champaign, IL 61801, USA; oelze@illinois.edu; 6Department of Radiation Oncology, Sciences Centre, Physical Sciences, University of Toronto, Toronto, ON M4N 3M5, Canada

**Keywords:** 3D breast acoustic CT, magnetic resonance imaging, breast masses

## Abstract

Breast cancer detection remains a significant challenge, with traditional mammography presenting barriers such as discomfort, radiation exposure, high false-positive rates, and financial burden. Moreover, younger women frequently fall outside routine mammographic screening guidelines, leaving critical gaps in early detection. **Objectives:** This study investigates the potential of quantitative transmission breast acoustic computed tomography scanner imaging (QT3D) as an innovative, non-invasive imaging modality for characterizing and evaluating breast masses. **Methods:** A comparative analysis between QT3D imaging and magnetic resonance imaging (MRI) was conducted in a cohort of patients with biopsy-proven benign or malignant breast lesions, comparing key metrics in quantifying breast masses for the purposes of breast mass characterization. **Results:** The findings in this study highlight its capability in identifying relatively small tumors, multiple lesions, satellite lesions, intraductal extensions, and calcifications, in addition to offering valuable diagnostic insights. **Conclusions:** This work is a first step toward studies essential for confirming its clinical feasibility, establishing its role in breast cancer tumor characterization, and potentially improving patient outcomes.

## 1. Introduction

Breast cancer is the most common non-epithelial malignancy diagnosed in women, with approximately 2.3 million new cases reported annually worldwide. In North America, it is projected that breast cancer will account for roughly 12% of all new cancer diagnoses and 6% of cancer-related deaths in 2025. It is estimated that 330,000 new cases will be diagnosed and 55,500 deaths will occur in North America, demonstrating the urgent need for improved detection and treatment approaches [1]. Accurate diagnosis and precise characterization of breast cancer are critical for effective treatment planning, enabling tailored therapeutic strategies that enhance patient outcomes. These statistics emphasize the necessity of advancing diagnostic technologies and intervention strategies to reduce the burden of breast cancer on individuals and healthcare systems [1,2].

Currently, the gold standard method of tumor diagnosis is pathological examination of core biopsy specimens. However, the invasive nature of the procedure required to collect core biopsies can cause post-surgical complications. Some masses also require repeat biopsy due to sampling errors during the initial biopsy. X-ray mammography and two-dimensional ultrasound B-mode imaging, two clinical imaging modalities that are used by radiologists for breast examination, provide limited information about the micro-structural properties of tissues [3,4]. There is a need for a three-dimensional non-invasive imaging modality that can provide rapid quantitative information that can be used to assist with breast tumor characterization.

Accurate imaging is essential in the management of breast cancer, guiding diagnosis, treatment planning, and disease monitoring [5]. Imaging modalities must provide reliable assessments of tumor size, morphology, and progression to optimize therapeutic strategies and improve clinical decision making. Mammography and handheld ultrasound remain integral components of breast imaging, particularly for initial screenings and supplemental evaluations [6]. Whereas mammography is highly effective in detecting microcalcifications and early-stage malignancies, it has limitations in imaging dense breast tissue and in accurately distinguishing between benign and malignant lesions, sometimes leading to false positives or missed diagnoses [7,8]. Ultrasound offers a dynamic, real-time assessment of breast lesions and is particularly useful for guiding biopsies, but it is operator-dependent, lacks standardization, and may struggle with differentiating tumor boundaries, leading to variability in interpretation and measurement accuracy [9,10].

Magnetic resonance imaging (MRI) is currently the most sensitive and widely utilized secondary modality for the detection, characterization, and treatment response assessment of breast malignancies. MRI leverages multiple imaging sequences, including T1-weighted, T2-weighted, and dynamic contrast-enhanced (DCE) imaging, to provide high-resolution anatomical detail and functional information regarding tumor vascularity and permeability [11]. Furthermore, diffusion-weighted imaging (DWI) and apparent diffusion coefficient (ADC) mapping offer insight into tissue cellularity and tumor microstructure [12]. Whereas MRI excels in sensitivity, its limitations include high operational costs, interobserver variability in lesion segmentation and enhancement kinetics subjectivity in interpretation, and limited accessibility in resource-constrained settings [13]. Additionally, MRI contrast agents, such as gadolinium-based compounds, pose potential risks for nephrotoxicity and long-term tissue deposition [14]. These constraints underscore the necessity for alternative imaging modalities that can deliver accurate, reproducible, and cost-effective breast cancer assessment while maintaining high diagnostic performance.

More recently, automated breast ultrasound has been increasingly explored as a supplemental imaging tool for breast cancer detection and response evaluation in patients. ABUS offers operator-independent, standardized imaging, addressing some of the limitations associated with handheld ultrasound, including variability in image acquisition and interpretation, and has been formulated in a number of different configurations [15,16,17]. Automated breast ultrasound has demonstrated sensitivity and specificity comparable to MRI in evaluating response to therapy. However, despite these advantages, automation has notable limitations in differentiating therapy-induced fibrosis from hypoechoic residual tumor tissue. Tumor fragmentation and stromal retraction can complicate interpretation, increasing the likelihood of false-positive findings. Additionally, while automated breast ultrasound improves lesion detection and measurement consistency, its diagnostic accuracy in predicting residual disease remains an area of active investigation.

Quantitative transmission tomographic imaging (QT3D) has emerged as a promising modality in this context, offering high-resolution, tomographic visualization of breast architecture through the analysis of transmitted ultrasonic waves. Unlike conventional reflection-based ultrasound modalities, quantitative ultrasound computed tomographic technology uses the accurate modeling of the propagation of acoustic waves through soft tissues or other methods to generate quantitative maps of tissue composition, displaying variations in acoustic impedance, speed of sound (SOS), and acoustic attenuation. These quantitative parameters facilitate precise tissue characterization and can enable the differentiation of malignant and benign lesions based on their acoustic properties [18,19,20,21].

One of the key advantages of breast ultrasound tomography is its ability to provide both image-based and volumetric tumor measurements with high spatial resolution. By utilizing computational tomography (CT)-like reconstructions, breast ultrasound tomography can be used to overcome the limitations of conventional two-dimensional B-mode ultrasound, which can be operator-dependent and lacks volumetric standardization. Preliminary studies suggest that tomographic ultrasound imaging demonstrates a significantly improved ability compared to handheld US in accurately measuring complex-shaped lesions [21,22]. This can be particularly relevant in the assessment of irregularly contoured or multifocal tumors, where conventional US is affected by interobserver variability and inadequate spatial resolution. The quantitative nature of ultrasound breast tomography with special measurements of speed of sound and attenuation allows for precise delineation of tumor margins and volumetric assessment, which is essential for tracking morphological changes and therapeutic response in patients undergoing oncology therapy. Additionally, because quantitative ultrasound computed tomographic imaging of the breast does not rely on exogenous contrast agents or ionizing radiation, it presents a safer and more accessible alternative to MRI, particularly for patients with contraindications to gadolinium-based contrast media. This makes this type of imaging amenable to use in a number of breast-imaging applications with no contrast-agent limitations nor radiation dose limitations.

This study aimed to systematically evaluate the efficacy of tomographic breast ultrasound imaging in breast cancer imaging, focusing on its ability to delineate breast mass morphology, quantify mass dimensions, and breast tumor characteristics. By comparing breast ultrasound tomography to MRI in the context of quantifying lesions for response assessment, this research seeks to establish a foundation for its potential integration into clinical practice, optimizing imaging strategies to enhance early detection; improve differentiation between benign and malignant lesions; and provide a non-invasive, cost-effective, and alternative approach for ongoing disease detection, characterization, and surveillance.

## 2. Materials and Methods

### 2.1. Study Design and Participants

This prospective, single-center, observational, early validation study aimed to evaluate the use of quantitative transmission tomographic imaging (QT3D) in detecting, characterizing, and measuring the volume of breast masses in comparison to breast MRI. The study was conducted at the Odette Cancer Centre, Sunnybrook Health Sciences Centre, Toronto, Canada. A total of 20 patients with suspected breast cancer were enrolled based on predefined eligibility criteria.

All participants provided written informed consent before study participation, adhering to good clinical practice guidelines and the principles outlined in the Helsinki Declaration. Clinical and demographic data, including tumor characteristics and treatment details, were collected from electronic medical records. The study protocol was approved by the Institutional Research Ethics Board (REB #5587) at Sunnybrook Health Sciences Centre and registered on ClinicalTrials.gov (NCT06175078) on 8 April 2024.

This observational study did not alter patient treatment plans. Data collected focused exclusively on ultrasound-based imaging for breast cancer characterization. Findings from QT3D were analyzed and compared to standard-of-care breast MRI, as well as histopathological results from routine core biopsy specimens, surgical reports, and radiological assessments available in patient charts.

### 2.2. Image Acquisition

#### QT Imaging (QT3D) Acquisition

Breast imaging was performed using a breast ultrasound computed tomographic system (QT Imaging Holdings, Inc., Novato, CA, USA). This device utilizes a 3D reflection mode and transmission mode imaging to provide quantitative tomographic ultrasound data [18]. Transmission imaging enables precise tissue characterization through speed of sound (SOS), as well as acoustic attenuation mapping, evaluated here in characterizing different breast tissue types.

During imaging, the patient was positioned prone on an ultrasound examination couch with one breast comfortably suspended in a water-filled scan tank maintained at 31 degrees Celsius. The scanning duration varied from 10 to 20 min, depending on breast size. The QT breast scanner 2000, Model A, and external components involved in this study are summarized in Figure 1.

The BACT system integrated a transmitter/receiver array pair and a reflection transceiver system and permitted simultaneous acquisition of transmission and reflection data. The transmitter–receiver pair is used exclusively for SOS/attenuation imaging, while three separate transceivers within the scan tank rotated around the breast to acquire reflection-mode ultrasound signals for 3D refraction-corrected imaging. These reflection arrays were tilted upwards at 10 degrees to optimize imaging near the chest wall. The imaging process involved capturing transmission data at 180 angles around the breast, interleaved with reflection mode data acquisition at a given level, moving vertically 2 mm, and repeating the process until the breast was scanned. Image reconstruction was performed using proprietary algorithms [19], which reconstructed quantitative variations in the acoustic properties across the whole breast at a mm resolution, to differentiate tissue types. The refraction-corrected reflection image used the SOS image and was compounded over 360 degrees, thus removing speckle and yielding a sub-mm resolution. This image was afterward automatically correlated with the SOS image.

Three-dimensional image sets were acquired with 1 cm transverse spacing along the longest axis of each tumor, centering the transducer over the estimated tumor midpoint. For comparative analysis, a corresponding normal tissue image was collected adjacent to the tumor to serve as an internal control. All imaging data were digitally archived and anonymized using unique non-identifiable patient codes to ensure privacy [20,21,22].

Tumor volume segmentation was performed using custom software. An initial ellipsoid was manually positioned to encompass the region of interest (ROI), and segmentation was refined using an ITK-based seed-based region-growing algorithm guided by the SOS value of the seed point. This method facilitated the delineation of tumor boundaries and improved volumetric assessment reliability [20,21,22]. The three fundamental imaging parameters are the speed of sound (SOS), which provides quantitative information about tissue stiffness and density, enabling tissue characterization; attenuation of sound, which helps identify regions with different absorption properties, often linked to tissue composition or pathological changes; and reflection, which offers structural detail at tissue interfaces and boundaries, aiding in anatomical visualization. We also point out that conventional 2D ultrasound forms images using reflection (backscatter) methodology.

This was an observational study alone, and patient treatments were not changed based on findings. The study only collected and analyzed ultrasound data with the aim of characterizing suspected breast cancers. The results of ultrasound data analysis were compared and correlated with the histopathology reports on routine core biopsy specimens, surgery reports, or radiology reports, which were available from patient charts.

Analyses of reflection images, speed of sound, attenuation, and values for masses were carried out. Analysis for statistical significance used the Wilcoxon rank-sum test.

### 2.3. Magnetic Resonance Imaging (MRI) Acquisition

As part of the comparative analysis, all patients underwent standard clinical breast MRI for tumor evaluation. MRI imaging was performed using a 1.5 T GE magnet and a dedicated breast coil as part of the patient’s standard of care. Localizer and bilateral T2, T1 non-fat sat, and T1 pre- and four to six dynamic runs post-gadolinium contrast images were obtained; 0.1 mmol/kg of IV gadolinium contrast was administered as a bolus during imaging. Axial T1-weighted fat-saturated 3D images were obtained. Post-processing imaging included subtraction imaging and MIP generation. Three-dimensional Reformatted MIP images were obtained for visualization. In addition, MRI data were used as a reference.

### 2.4. Comparative Analysis

MRI data were used as a reference standard for tumor size and morphology. Tumor segmentation on MRI was performed using semi-automated software (3D Slicer, www.slicer.org), with volumetric measurements compared against QT3D-derived metrics to evaluate accuracy and concordance between modalities. Quantitative parameters from QT3D, specifically linear measurements and lesion volume as assessed in speed-of-sound images, were compared to MRI-based tumor assessments. Tumor volume and response measurements were analyzed to determine both correlation and absolute agreement using Bland–Altman analysis with irregular ellipsoid assumptions. Correlation analysis was conducted to assess the trend in bias in size measurements, while the Bland–Altman method was used to assess and quantify that bias. All statistical analyses were conducted using MATLAB 2025 (Natick, MA, USA).

## 3. Results

### 3.1. Patient and Tumor Characteristics

A total of 20 female patients were enrolled at the Louise Temerty Breast Cancer Centre and Sunnybrook Health Sciences Centre, presenting with 16 primary breast tumors and 4 benign lesions. As part of their standard care, all patients underwent a core needle biopsy prior to treatment to confirm the cancer diagnosis; determine the histological subtype; and assess hormone receptor status, including estrogen receptor (ER), progesterone receptor (PR), and human epidermal growth factor receptor (HER2) in order to classify the tumor’s molecular subtype. Pre-treatment magnetic resonance imaging (MRI) was performed as part of clinical care to establish the initial tumor size. The clinical and pathological characteristics of the patients involved in this study are summarized in Table 1. The mean patient age was 54 years (range 24–76). There were 16/20 patients with tumors and 4/20 patients with benign disease. For patients with malignant disease, 1/16 patients had grade 1 tumors, 8/16 had grade II tumors, and 7/16 had grade III tumors; 12/16 of patients with tumors were ER+, 11/16 were PR+, and 3/16 were ER-, PR-, and Her2-. The most common histopathological tumor type was invasive ductal carcinoma.

### 3.2. Quantitative and Morphological Features in 3D Reconstruction

Image data were used to directly determine automated volumetric breast morphological features in 3D and for lesion volume calculations derived from speed of sound tissue maps. For this study, the breast masses’ characterization, extension, calcification, and individual volumes were calculated and described for each lesion.

Mass characterization was dependent on 3D evaluation of the breast visualized using different fundamental images: speed of sound (SOS), attenuation of sound, and reflection images. Estimation of the volume of the target mass was performed using 3D images in the sagittal, axial, and coronal planes from speed of sound images (Figure 2 and Figure 3). Most benign lesions appeared well circumscribed and were hypoechoic on reflection imaging. They had a well-defined appearance on speed of sound imaging and appeared prominent in attenuation images. In contrast, for malignant lesions, a spiculated appearance was most typical, with again obvious features on speed of sound imaging and less obvious features on attenuation images.

In terms of quantitative measures, malignant tumors exhibited a mean speed of sound of 1554 ± 27 m/s and an attenuation of 29 ± 3 dB/m. Benign masses exhibited a speed of sound of 1499 ± 73 m/s and an attenuation of 16 ± 7 dB/m. Reflection values indicated values of 17 ± 5 dB for benign masses versus 24 ± 6 dB for malignant cases.

Measurements were made from regions of interest encompassing only masses of breast abnormality. Of the examined metrics, statistically significant differences in values were only demonstrated for the minimum attenuation and the average attenuation of masses (at the 5% significance level) (Table 2).

A comparative analysis of the morphological characteristics of masses for patients involved in this study is summarized in Table 3. A summary of the comparative analysis of biopsied mass measurements in study patients is presented in Table 4. In MRI data, the mean breast mass dimensions were 3.4 ± 1.6 cm, whereas in the 3D QT images, they were 2.9 ± 1.0 cm. In terms of mass volume, from MRI data, the mean breast mass volume was 19.4 ± 22.9 cm^3^, whereas in the 3D QT images it was 12.4 ± 10.3 cm^3^. Results indicated good concordance with the context that contrast-based MRI imaging is expected to identify areas of non-mass contrast uptake that may not be apparent in breast mass-based imaging. Mammographic images were obtained using standard 2D full-field digital mammography systems as part of the patient’s standard of care. 

A schematic diagram with comparative image results obtained using 3D ultrasound tomography, 2D mammography and magnetic resonance imaging for the characterization of breast masses is attached in Figure 4.

The Bland–Altman method, a technique for assessing agreement between two methods of clinical measurement by analyzing the differences between paired measurements, was used to assess the level of agreement between tumor volume estimates made from QT3D ultrasound images and MRI images (Figure 5 and Figure 6). It was evident that for 15 primary breast cancer cases, the volumetric differences between MRI and QT3D scans generally fell within the limits of agreement, signifying consistent measurement alignment between the two image modalities. Only one case fell outside these limits. This outlier demonstrated a significant discrepancy in the volume measurements, suggesting that the two methods yielded substantially different results for this breast mass (Patient Case #2, with a complex image represented as a conglomerate mass and in MRI non-mass enhancement involving all four quadrants).

## 4. Discussion

This study presents initial findings from a cohort of 20 patients enrolled at Sunnybrook Health Sciences Centre in Toronto, Canada, who underwent breast ultrasound tomographic imaging. This cohort comprised 16 patients with primary breast tumors and 4 with benign masses.

As expected, the average age at diagnosis among the cohort was 53.5 years, with the youngest patient diagnosed at 24 years and the oldest at 76 years, which is typical for breast cancer diagnoses. As also expected, breast masses were symmetrically distributed, with 50% of cases involving the right breast and 50% the left breast. Radiological assessment using the BI-RADS classification revealed that 60% of lesions were categorized as BI-RADS 5, indicating a high likelihood of malignancy, 30% as BI-RADS 4, and 10% as BI-RADS 0.

Histological evaluation identified invasive ductal carcinoma as the most prevalent subtype, as expected, accounting for 11 cases (69%), followed by invasive lobular carcinoma in 2 cases (13%), and mixed histology in 3 cases (18%). Grading of the tumors revealed that 50% were Grade II, 43% were Grade III, and 7% were Grade I, reflecting a predominance of moderately to poorly differentiated tumors.

Molecular subtyping was also as expected and was based on biomarker analysis, demonstrating a diverse profile. Six cases (38%) were classified as hormone receptor-positive (Luminal), aligning with favorable prognostic features. Five cases (32%) were triple-positive, expressing estrogen receptor (ER), progesterone receptor (PR), and HER2. Four cases (25%) were triple-negative. Notably, one case (5%) was HER2-positive only, a less common but clinically significant subtype given the availability of targeted therapies. The number of patients was too small to conduct a subgroup analysis regarding quantitative values of speed of sound and attenuation, which appeared to be only different between the malignant and benign groups of breast masses.

Standard magnetic resonance imaging (MRI) plays a role in the evaluation of breast cancer, offering precise volumetric assessment and detailed characterization of breast masses. It is essential for determining optimal oncological and surgical treatment strategies and for monitoring therapeutic responses over time. The objective of the work here was to identify optimal quantitative 3D ultrasound (QT3D) parameters that can non-invasively and efficiently characterize breast masses. Unlike MRI, QT3D requires no contrast agents or radiation, making it a safer and more accessible alternative. The diagnostic performance of QT3D was rigorously validated against histopathological findings, with the goal of achieving accuracy equal to or greater than that of MRI.

As expected, the speed of sound values for masses were fairly consistent, with no difference noted for the benign versus malignant masses that was statistically significant (in the context of a limited patient population, n = 20). In contrast, as expected, attenuation values were different between malignant and benign masses. Specifically, the malignant and benign mass tumors exhibited comparable speed of sound but differing attenuation and reflection values. This was expected and consistent with other studies, mainly linked to studies of ultrasound backscatter properties, which are different between benign and malignant breast masses.

The characterization of breast masses was conducted using both MRI and QT3D imaging techniques. QT3D imaging identified tumors measuring 2–5 cm (T2 classification) in 88% of cases (14 patients), while in 12% of cases (2 patients), the tumors were smaller than 2 cm, falling under the T1 classification. Notably, QT3D imaging revealed multiple lesions in 68% of cases, whereas 32% (5 cases) involved a single lesion. Additionally, QT3D imaging facilitated the detection of satellite lesions, intraductal extensions, posterior extensions toward the pectoral muscle, and calcifications—all without the need for contrast enhancement. However, MRI identified tumors measuring between 2 and 5 cm (T2) in 50% of cases, tumors larger than 5 cm (T3) in 38% of cases, and smaller lesions less than 2 cm (T1) in 12% of cases (2 patients). MRI detected multiple lesions in 62% of cases, while 38% of cases involved single, non-complex lesions.

The observed discrepancy between MRI and QT3D scan measurements may stem from several factors:(i)Image Representation Variability: Differences in how each modality visualizes breast tissue and defines mass boundaries can lead to inconsistent volume estimates.(ii)Breast Density: Variations in tissue composition can influence image quality and segmentation accuracy, especially in dense breasts.(iii)MRI Contrast: The uptake and distribution of contrast agents (perfusion) may affect the delineation of mass margins.(iv)Patient-Specific Anatomical Factors: Individual variations, such as mass shape, location, or proximity to anatomical structures, could pose challenges for measurement consistency.(v)Technical Differences in Imaging Modalities: Disparities in resolution, field of view, and algorithms for capturing and processing volumetric data may contribute to measurement variability.

These findings underscore key differences in mass detection and characterization between the two imaging modalities. Whereas QT3D demonstrated good performance in identifying tumors classified as T2 and detecting multiple lesions, MRI offered a broader distribution in size classification, including a notable proportion of tumors in the T2–T3 category. This suggests that QT3D could excel in specific diagnostic scenarios, particularly when evaluating advanced or multifocal disease.

The Bland–Altman plot indicated a “good overall agreement between MRI and 3D QT scan for assessing breast mass volume in most cases”. However, outlier presence is most likely due to non-mass enhancement on MRI with contrast, underscoring an area for further investigation and potential optimization. Future steps could include reviewing imaging protocols and segmentation techniques, as well as conducting sensitivity analyses to determine the impact of breast density and contrast use and implementing quality assurance measures to minimize variability in volumetric measurements. Such refinements would enhance the reliability and clinical utility of these imaging modalities in breast mass assessment.

## 5. Conclusions

In conclusion, quantitative ultrasound breast tomography represents a significant advancement in breast cancer imaging, offering unique advantages that can complement existing modalities like MRI. It has been integrated into the diagnostic workflow and has the potential to enhance abnormal breast mass characterization, optimize treatment planning, and potentially improve patient outcomes.

Quantitative ultrasound tomography demonstrated diagnostic performance comparable to that of MRI in this study population. Tomographic ultrasound may serve as an effective alternative in scenarios where MRI is not feasible due to availability or patient contraindications.

## Figures and Tables

**Figure 1 tomography-11-00111-f001:**
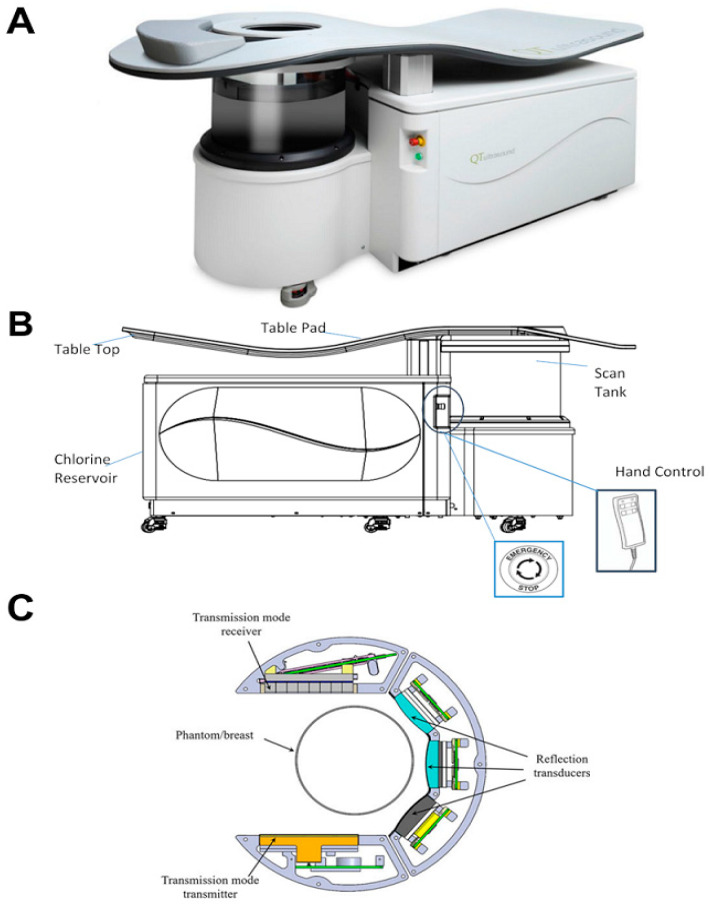
Schematic of the QT3D breast tomographic device used. (**A**) Device image, (**B**) schematic of transducers and receivers, and (**C**) device schematic.

**Figure 2 tomography-11-00111-f002:**
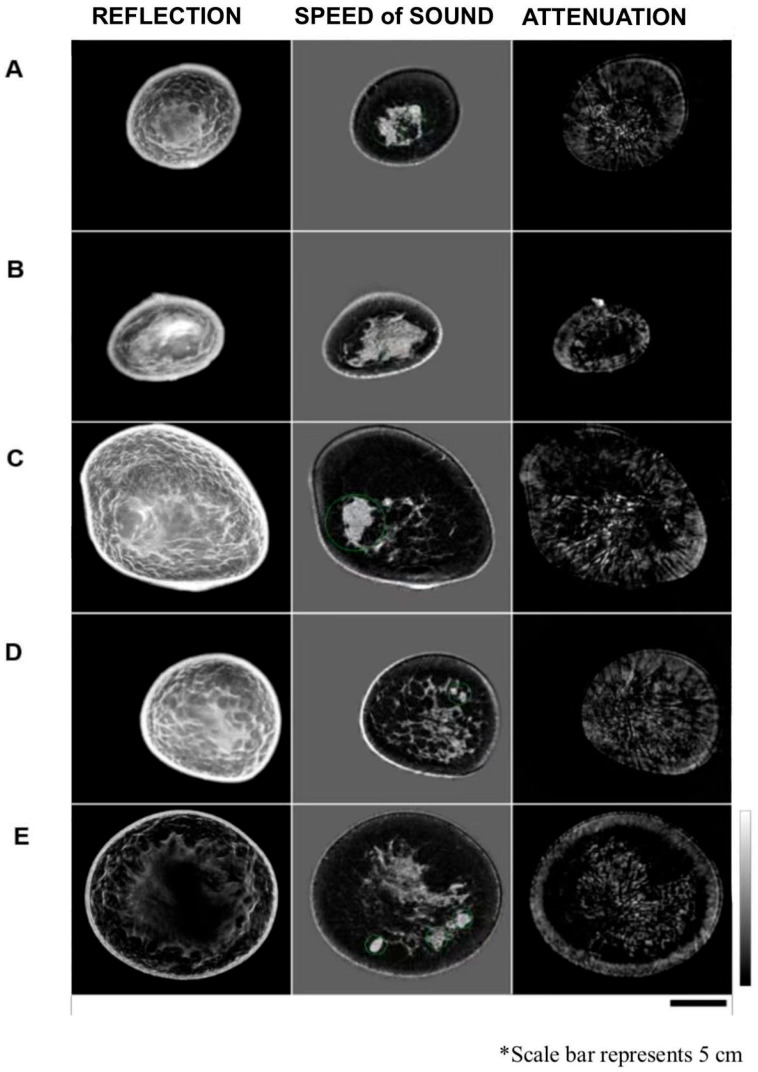
Three-dimensional reflection and quantitative images and morphological features in 3D reconstruction. (**A**): ILC, left breast, multicentric lesions, UOQ, and LIQ. (**B**): IDC, right breast, unifocal, LIQ. (**C**): Ductal and metaplastic carcinoma, right breast, unifocal, and UOQ. (**D**): IDC, Left breast, multifocal. (**E**): Multiple benign lesions, left breast, LOQ, and LIQ. The vertical gray scale bar indicates measures of 1360 m/s to 1640 m/s for speed of sound images, and 0 dB/m to 500 dB/m for attenuation images. For reflection images, the gray scale bars indicate 0 to 42 dB. Scale bar represents 5 cm.

**Figure 3 tomography-11-00111-f003:**
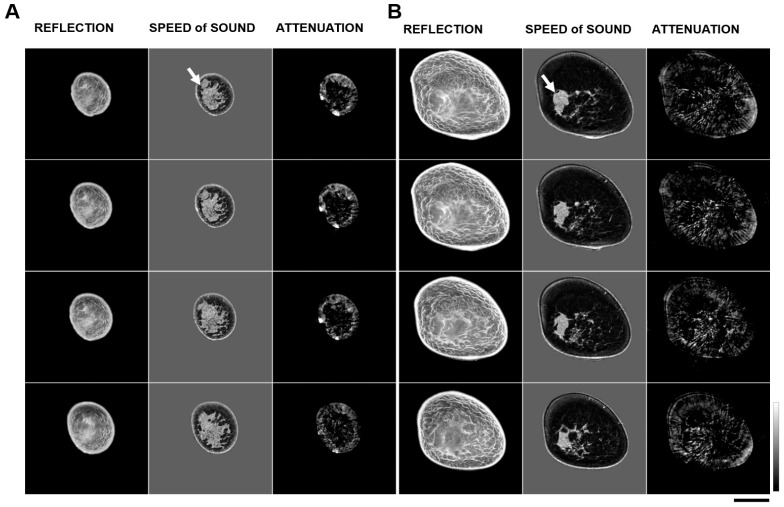
Representative tomographic images. Volumetric images are shown for a (**A**) benign breast tumor and a (**B**) malignant breast tumor. Representative slices are from distal to proximal through the breast in a prone position with the breast pendant. The vertical gray scale bar indicates measures of 1360 m/s to 1640 m/s for speed of sound images, and 0 dB/m to 500 dB/m for attenuation images. For reflection images, the gray scale bars indicate 0 to 42 dB. Scale bar represents 5 cm. Arrows (top slice only—attenuation images) indicate the position of breast mass.

**Figure 4 tomography-11-00111-f004:**
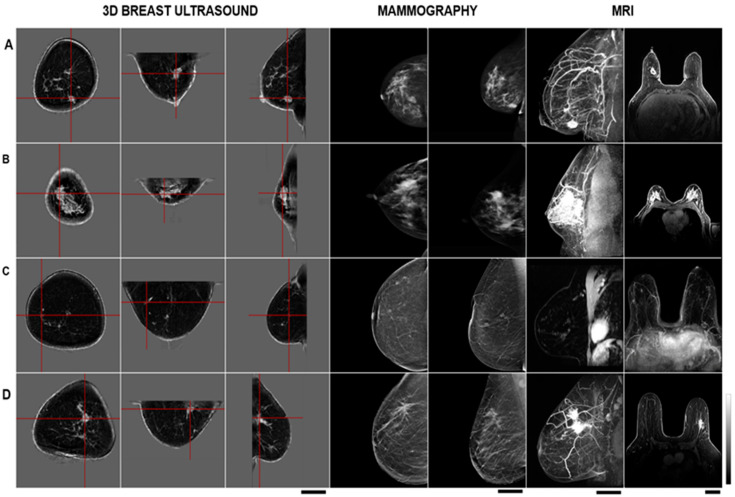
The schematic diagrams alongside comparative imaging results obtained from 3D ultrasound tomography, 2D mammography, and MRI for the characterization of breast masses. (**A**): IDC, right breast, unifocal, LIQ, multiple 3D images using ultrasound tomography, mammography, and MRI. (**B**): IDC, right breast, unifocal, UOQ, multiple 3D images using ultrasound tomography, mammography, and MRI. (**C**): Multifocal recurrence breast cancer, IDC, right side, UOQ, multiple 3D images using ultrasound tomography, 2D mammogram, and MRI. (**D**): IDC, left breast, multifocal, and UOQ, with multiple 3D images using ultrasound tomography, mammography, and MRI. Scale bar represents 5 cm.

**Figure 5 tomography-11-00111-f005:**
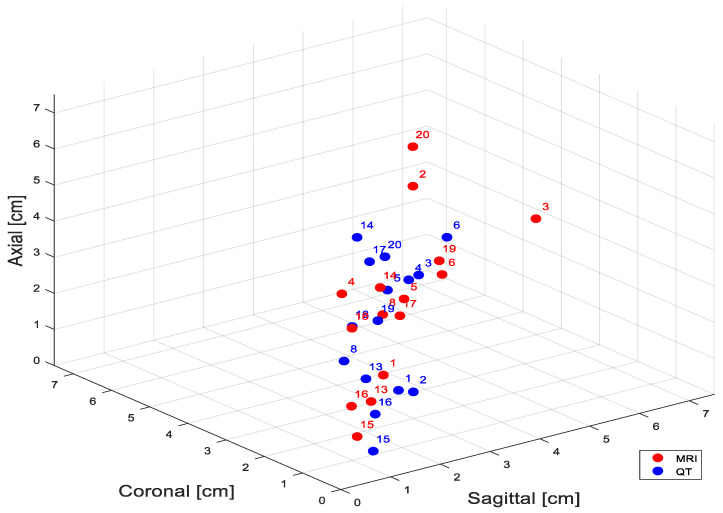
Comparative size measurements between masses characterized by quantitative ultrasound breast CT versus MRI. Blue circles indicate ultrasound measurements versus red for MRI-based measurements.

**Figure 6 tomography-11-00111-f006:**
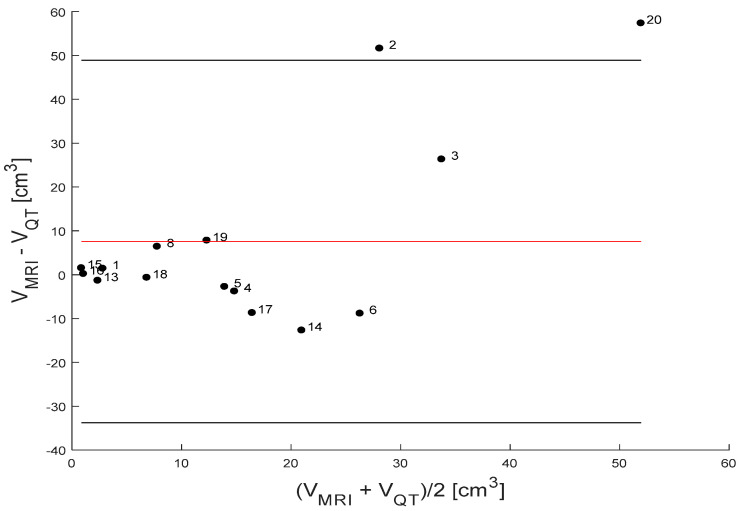
Bland Altman plot comparing mean tumor volume estimates and differences between MRI and QT imaging measurements. The red line in the plot represents the mean difference (or bias) between the two methods, providing an overall estimate of their agreement. The two black lines indicate the limits of agreement, defined as ±1.96 times the standard deviations from the mean difference. These limits help determine the range within which most differences are expected to fall, assuming a normal distribution of data.

**Table 1 tomography-11-00111-t001:** Patient demographics and characteristics.

Case	Age	Breast	BIRADS	Pathology	Grade	Estrogen Receptor	Progesterone Receptor	Her2 Receptor
01	54	Left	5	IDC	2	+	+	-
02	70	Left	5	ILC	2	+	+	-
03	51	Left	5	IDC	2	-	-	-
04	56	Right	4c	IDC	3	+ (Low)	-	+
05	45	Right	0	IDC	3	+	+	-
06	48	Right	5	IDC/Metaplasic	2	+	+	+
07	45	Left	5	IDC	2	+	+	+
08	76	Left	5	IDC/Lobular/Mucinous	1	+	+	-
09	31	Left	4	Fibroepithelial Lesion				
10	60	Right	0	Focal Apocrine Metaplasia				
11	24	Right	4	Fibroepithelial Lesion				
12	50	Right	4	Fibroepithelial Lesion				
13	56	Right	5	IDC	3	-	-	-
14	61	Left	5	IDC	3	+	+	-
15	55	Right	4c	IDC/Recurrence	2	+	+	+
16	73	Right	5	ILC	2	+	+	-
17	43	Left	5	IDC/Lobular	2	+	+	+
18	47	Right	5	IDC	3	+	+	+
19	49	Left	4c	IDC	3	-	-	+
20	67	Left	5	IDC	3	-	-	-

* IDC: invasive ductal carcinoma, ILC: invasive lobular carcinoma.

**Table 2 tomography-11-00111-t002:** SOS, attenuation, and reflection values for breast masses.

	Mean (Malignant)	St. Dev. (Malignant)	Mean (Benign)	St. Dev. (Benign)	*p*-Value
Minimum Speed of Sound (m/s)	1518	32	1462	73	0.276
Maximum Speed of Sound (m/s)	1614	19	1589	41	0.369
Average Speed of Sound (m/s)	1554	27	1499	73	0.277
Minimum Attenuation (dB/m)	14	3	5	4	0.00392
Maximum Attenuation (dB/m)	51	6	40	15	0.219
Average Attenuation (dB/m)	29	3	16	6	0.00289
Minimum Reflection (dB)	19	5	14	6	0.119
Maximum Reflection (dB)	31	9	21	5	0.0526
Average Reflection (dB)	24	6	17	5	0.0470

**Table 3 tomography-11-00111-t003:** Comparative analysis of 2D mammography, MRI, and QT3D for breast mass characterization.

Case	Mx Description	MRI Description	QT3D Description
**01**	Irregular noncalcified mass at 9:00 In left breast.	Enhancing mass with multiples satellite Nodularity in UIQ.	Multiple, irregular, spiculated dense masses with architectural distortion and calcification.
**02**	Irregular, lobulated dense mass subareolar, left breast.	Conglomerate mass and non-mass enhancement involving all 4 quadrants.	Multicentric, irregular, dense masses associated with calcification and nipple extension.
**03**	Large asymmetrical density in left breast UOQ.	Large area of conglomerate mass and non-mass enhancement in UOQ.	Large central area, irregular dense mass posteriorly with pectoralis muscle extension.
**04**	Hypoechoic solid mass retroareolar In right breast.	Right central lower mass enhancement. There is no mass enhancement inferiorly with nipple areola extension.	Large central area, irregular dense mass with nipple extension associated with calcification.
**05**	There is a global asymmetry in right breast, more prominent in central and medial.	Irregular enhancing mass in LOQ with pectoralis muscle and nipple extension.	Large and irregular dense mass with nipple, skin, and muscle extension; Calcifications in central area.
**06**	Dense ill-defined mass associated with calcification in right breast UOQ and medial.	Heterogeneously enhancing mass in UOQ with 2 satellite lesions.	Irregular dense mass associated with calcification. Satellite lesion with calcification at 9:00.
**07**	Increasing pleomorphic Microcalcifications in UOQ left breast, which have linear distribution.	Several new enhancing masses without bright T2 signal with mixed persistent Washout kinetic in UOQ.	Four nodules irregular associated with calcification in UOQ. Satellite irregular lesion associated with calcification in UIQ.
**08**	Spiculated irregular dense mass In UOQ left breast.	Irregular and spiculated dense mass with increasing enhancement in UOQ and satellite lesion.	Spiculated, irregular, dense mass associated with calcification, and satellite lesion. Second spiculated irregular dense mass associated with calcification.
**13**	Irregular mass with spiculated Margins associate with microcalcification LIQ right breast.	Irregular spiculated mass with increasing enhancement in LIQ, 5:00. Another similar smaller irregular enhancing mass in 3:00.	Spiculated irregular dense mass associated with calcification. Second spiculated irregular dense mass associated with calcification.
**14**	Irregular dense microlobulated mass associated with calcifications; extend toward the nipple areolar complex in UIQ left breast.	Irregular heterogeneous mass with increasing enhancement at 11:00 to 3:00. Multiple satellite lesions around 10 mm.	Large and irregular dense mass with multiple satellite lesions. Surrounding associated with calcification.
**15**	New nodule right breast lateral LOQ.	Nodule right breast anterior to mid third central infer lateral nipple/scar. New nodule right breast lateral LOQ. Both nodules have increasing enhancement.	Two heterogeneous nodules.
**16**	New spiculated posterior mass in UOQ right breast.	Irregular mass with spiculated margins posterior in UOQ.	Spiculated, irregular, dense mass associated with calcification. Satellite lesion. Mammary duct ectasia retro nipple.
**17**	Architectural distortion in the left UOQ posterior.	Irregular spiculated mass with border heterogeneous in 1:00.	Spiculated, irregular, dense mass associated with calcification and posterior small satellite lesion.
**18**	Irregular spiculated mass containing irregular calcifications in UOQ right breast.	Irregular, ill-defined, and spiculated heterogeneously enhancing mass in UOQ. Posteriorly small satellite lesion.	Spiculated, irregular, dense masses associated with calcification and posterior extension.
**19**	At the left 12:00 breast anterior– middle third, area of focal asymmetry with associated microcalcifications.	Irregular, spiculated, heterogeneously enhancing mass within the left breast at 12:00; 1.2 cm from the nipple areolar complex.	Spiculated, irregular, dense mass associated with calcification. Two heterogeneous satellite nodes at 11:00–12:00, 10 mm. Irregular node associated with calcification at 8:00.
**20**	Irregular spiculated dense mass with associated architectural distortion in UOQ left breast.	Irregular spiculated border heterogeneous enhancing mass at 2:00, 5FN. Associated linear nonmass enhancement extending anterosuperiorly.	Spiculated, irregular, dense mass associated with calcification, posterior, with pectoralis muscle extension. Associated linear nonmass extending anterosuperiorly with calcification. Satellite lesion associated with calcification in central area.

**Table 4 tomography-11-00111-t004:** Comparative analysis of 2D mammography, MRI, and QT3D for breast mass sizes.

Case	Mx Size	MRI Size	QT3D Size
01	14 × 8 × 13 mm	24 × 20 × 14 mm Overall Disease 42 × 40 × 14 mm	22.2 × 13.7 × 13.3 mm 13.5 × 11.6 × 8 mm 11.5 × 9.4 × 9.7 mm
02	17 × 11 × 16 mm	67 × 54 × 37 mm	27.1 × 16.2 × 10 mm 11.6 × 10.6 × 12.6 mm
03	37 × 18 × 33 mm	70 × 40 × 32 mm	40.3 × 32 × 30.5 mm
04	21 × 16 × 24 mm	26 × 23 × 23 mm Overall Disease 52 × 23 × 23 mm	35 × 27.7 × 33 mm
05	32 mm	32 × 25 × 30 mm	32.5 × 30 × 30 mm
06	37 mm	45 × 32 × 29 mm Satellite Lesions 11 and 12 mm	46.6 × 32.8 × 38.4 mm Satellite Lesion 14 mm
07	40 mm	6 Nodules: 10, 8, 6, 7, 10, and 10 mm. Overall Disease 65 mm	4 Nodules: 13.7, 9.2, 6.8, and 10.1 mm Satellite Lesion 10 mm
08	23 × 17 × 16 mm	30 × 28 × 25 mm Satellite Lesion 8 mm	20.7 × 26 × 16.3 mm. Satellite Lesion 10.9 mm
13	19 × 16 × 9 mm	20 × 18 × 9 mm 12 × 7 × 4 mm	20.5 × 20 × 14.2 mm 10 × 9 × 6 mm
14	62 × 56 × 41 mm	36 × 35 × 37 mm Overall Disease 52 × 66 × 83 mm	33.3 × 39 × 40.2 mm Satellite Lesions 15.8 and 11 mm
15	7 × 7 × 6 mm	8 × 6 × 9 mm 7 × 6 × 6 mm	10 × 4.4 × 5 mm 10 × 4.8 × 4 mm
16	17 mm	13 × 14 × 12 mm	16 × 11.7 × 9.8 mm
17	28 × 24 × 22 mm	35 × 30 × 22 mm	32.2 × 34.3 × 36 mm
18	26 mm	20 × 23 × 27 mm Satellite Lesion 5 mm	21.2 × 24.5 × 26.4 Satellite Lesion 7 mm
19	14 × 11 × 13 mm	36 × 21 × 41 mm	25.5 × 23.4 × 27 mm Satellite Lesions 10 mm, 10 mm, and 10 × 7 × 6 mm
20	35 × 25 mm	60 × 48 × 29 mm Overall Disease 11 × 60 × 48 mm	35.6 × 34.7 × 36 mm with Anterior extension (overall 45.8 mm) Satellite Lesion 10 × 7 × 3 mm

## Data Availability

The data presented in this study are available upon request from the corresponding author in accordance with the institutional policies of Sunnybrook Research Institute.

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
