# Peer review of "Quantitative Volumetric Analysis Using 3D Ultrasound Tomography for Breast Mass Characterization"

_tomography, 2025, doi:10.3390/tomography11100111_

Round 1

Reviewer 1 Report

Comments and Suggestions for Authors

Comments

The article describes the uses of Quantitative Transmission Tomographic Imaging (QT3D). The results are presented neatly and justified against the gold standard MRI. Reviewers have few concerns and suggestions regarding the content and method which could be answered by the authors.

  • It would be interesting to know some of the image reconstruction methods’ names at this place. As it is not apparent from the reference used.

“Image reconstruction was performed using proprietary algorithms (19), which reconstructed quantitative variations in the acoustic properties across the whole breast at an mm resolution, to differentiate tissue types.”

  • Figure 3 is distorted by stretching it only in one direction ( Horizontal). It must be redrawn.
  • Three different fundamental imaging must be explained a little more in the text which are speed of sound (SOS), attenuation of sound and reflection. These are the principals of image generation and must be highlighted properly.
  • Reviewer’s major concern is the dimension measured by QT3D and gold standard MRI methods. The mean value of dimensions measured by both methods have significant differences.
  • Also, is this the largest dimension measured in one direction? It should be mentioned in the text.
  • Seed based region growing method has limitation of manual seed point selection. How did authors handled this situation? How can they overcome it?

Author Response

Comments 1: The article describes the uses of Quantitative Transmission Tomographic Imaging (QT3D). The results are presented neatly and justified against the gold standard MRI. Reviewers have few concerns and suggestions regarding the content and method which could be answered by the authors.

It would be interesting to know some of the image reconstruction methods’ names at this place. As it is not apparent from the reference used.

 “Image reconstruction was performed using proprietary algorithms (19), which reconstructed quantitative variations in the acoustic properties across the whole breast at an mm resolution, to differentiate tissue types.”

Figure 3 is distorted by stretching it only in one direction ( Horizontal). It must be redrawn.

Three different fundamental imaging must be explained a little more in the text which are speed of sound (SOS), attenuation of sound and reflection. These are the principals of image generation and must be highlighted properly.

Reviewer’s major concern is the dimension measured by QT3D and gold standard MRI methods. The mean value of dimensions measured by both methods have significant differences.

Also, is this the largest dimension measured in one direction? It should be mentioned in the text.

Seed based region growing method has limitation of manual seed point selection. How did authors handled this situation? How can they overcome it?

Response 1:

  • “It would be interesting to know some of the image reconstruction methods’ names at this place. As it is not apparent from the reference used”: We have ensured that the reference provided provides the methodological and mathematical details pertaining to the tomographic methods used.
  • “Figure 3 is distorted by stretching it only in one direction (Horizontal). It must be redrawn.”: We agree with the reviewer’s observation. Figure 3 has been redrawn to maintain the correct aspect ratio and eliminate horizontal distortion.
  • “Three different fundamental imaging must be explained a little more in the text which are speed of sound (SOS), attenuation of sound and reflection. These are the principles of image generation and must be highlighted properly.”: We have clarified the description of the three fundamental imaging parameters in the revised manuscript (Line 194 - 200): Speed of Sound (SOS): Provides quantitative information about tissue stiffness and density, enabling tissue characterization. Attenuation of Sound: Helps identify regions with different absorption properties, often linked to tissue composition or pathological changes. Reflection: Offers structural detail at tissue interfaces and boundaries, aiding in anatomical visualization. We also point out that conventional 2D ultrasound forms images using reflection (backscatter) methodology.
  • “Reviewer’s major concern is the dimension measured by QT3D and gold standard MRI methods. The mean value of dimensions measured by both methods have significant differences.” We now annotate in discussion that these differences are expected and may stem from the inherent modality-specific contrasts and resolution, as well as patient positioning during the scans. This difference may lead to slight variations in lesion shape or size. It is also well appreciated that non-contrast changes in MRI may also manifest differently in ultrasound imaging (on backscatter) although effects on attenuation and speed of sound are not yet well appreciated.
  • “Also, is this the largest dimension measured in one direction? It should be mentioned in the text.”: Yes, the dimensions reported correspond to the largest diameter of the lesion measured in any direction (i.e., the maximum axial, sagittal, or coronal dimension, depending on lesion orientation).
  • “Seed based region growing method has limitation of manual seed point selection. How did authors handle this situation? How can they overcome it?”: Readers at QT3D manual seed point selection by implementing a semi-automated seed selection process based on predefined criteria. This approach reduces user dependency and enhances the reproducibility of the region growing method. However, while this semi-automated method alleviates some of the challenges associated with manual seed placement, future work could focus on developing fully automated seed selection techniques. Such advancements would further improve the method’s robustness, consistency, and applicability across diverse datasets.

Reviewer 2 Report

Comments and Suggestions for Authors

General Comments

This is an important and timely paper that seeks to compare quantitative transmission 3D breast ultrasound (QT3D) with magnetic resonance imaging (MRI) for the characterisation of breast lesions. That said, I do have a few issues which I would like to see the authors address.

First, I was surprised that the abstract makes very little, if any, mention of the patients who were studied. Obviously this was an "enriched" population with all patients having either malignant or benign lesions. 

Second, the authors correctly point out that QT3D does not have the problem of ionising radiation and breast compression suffered by mammography. They also point out that among the limitations of MRI include "high operational costs" and the use of contrast agents. However, I was surprised to discover in a recent report on AuntMinnie.com (https://bit.ly/45BRkIg) that the cost of a QT3D system is about $500,000, which would certainly limit its use in resource-constrained settings.

Third, there is extensive use of acronyms but on a few occasions these are not defined. I refer here to BACT in line 170, ITK in line 191, as well as UOQ and LIQ in line 265 (and elsewhere). I'd like to suggest that the authors add a table at the end of the manuscript where all the abbreviations are defined.

Fourth, the labels in Figure 1 are essentially illegible. Since three of the authors are from QT Imaging, surely you can present a better quality image?

Fifth, the aspect ratio of the breast cross-sections in Figure 2 seems to be distorted, with images "squashed" in a vertical direction. In contrast, Figure 3 looks to be okay.

Sixth, as the authors well know, not all automated breast ultrasound (ABUS) systems are the same. Whereas the systems from QT Imaging and Delphinus Medical Technologies implement both reflective and transmissive ultrasound waves, the widely deployed Invenia system manufactured by GE HealthCare uses only reflective ways. This needs to be emphasised in the manuscript.

Seventh, no mention is made in the methods about the acquisition of mammographic data and yet these are presented in Table 4 and Figure 4. I was also puzzled to understand if the mammography was 2D or 3D (i.e. digital breast tomosynthesis). The caption to Table 4 indicates "2D mammography", and yet the data in the Mx column suggests 3D measurements of the breast masses. Please clarify.

Finally, I was interested to see whether the authors might conclude by suggesting that QT3D could replace MRI (see lines 386 to 388) but they appeared to shy away from this conclusion. Is their recommendation that one approach is better than the other or are they essentially equivalent?

Specific Comments

line 108    ... of conventional two-dimensional B-mode ultrasound ...

line 184    A 1 cm transverse spacing seems quite large. Can the system not support 1 mm spacing?

line 213    MRI data were used as reference ...

line 248    ... characterization, extension and calcification, individual ...

line 308    Re-order the three imaging modalities so they match the order in the table

line 359    The average age at diagnosis among ...

line 408    The observed discrepancies between MRI and ...

line 451     ... National Institute of Health in  Canada under grant number ...

Author Response

Comments 2: This is an important and timely paper that seeks to compare quantitative transmission 3D breast ultrasound (QT3D) with magnetic resonance imaging (MRI) for the characterisation of breast lesions. That said, I do have a few issues which I would like to see the authors address.

First, I was surprised that the abstract makes very little, if any, mention of the patients who were studied. Obviously this was an "enriched" population with all patients having either malignant or benign lesions.

Second, the authors correctly point out that QT3D does not have the problem of ionising radiation and breast compression suffered by mammography. They also point out that among the limitations of MRI include "high operational costs" and the use of contrast agents. However, I was surprised to discover in a recent report on AuntMinnie.com (https://bit.ly/45BRkIg) that the cost of a QT3D system is about $500,000, which would certainly limit its use in resource-constrained settings.

Third, there is extensive use of acronyms but on a few occasions these are not defined. I refer here to BACT in line 170, ITK in line 191, as well as UOQ and LIQ in line 265 (and elsewhere). I'd like to suggest that the authors add a table at the end of the manuscript where all the abbreviations are defined.

Fourth, the labels in Figure 1 are essentially illegible. Since three of the authors are from QT Imaging, surely you can present a better quality image?

Fifth, the aspect ratio of the breast cross-sections in Figure 2 seems to be distorted, with images "squashed" in a vertical direction. In contrast, Figure 3 looks to be okay.

Sixth, as the authors well know, not all automated breast ultrasound (ABUS) systems are the same. Whereas the systems from QT Imaging and Delphinus Medical Technologies implement both reflective and transmissive ultrasound waves, the widely deployed Invenia system manufactured by GE HealthCare uses only reflective ways. This needs to be emphasised in the manuscript.

Seventh, no mention is made in the methods about the acquisition of mammographic data and yet these are presented in Table 4 and Figure 4. I was also puzzled to understand if the mammography was 2D or 3D (i.e. digital breast tomosynthesis). The caption to Table 4 indicates "2D mammography", and yet the data in the Mx column suggests 3D measurements of the breast masses. Please clarify.

Finally, I was interested to see whether the authors might conclude by suggesting that QT3D could replace MRI (see lines 386 to 388) but they appeared to shy away from this conclusion. Is their recommendation that one approach is better than the other is or are they essentially equivalent?

line 108    ... of conventional two-dimensional B-mode ultrasound ...

line 184    A 1 cm transverse spacing seems quite large. Can the system not support 1 mm spacing? Yes, the system can support 1mm spacing.

line 213    MRI data were used as reference ...

line 248    ... characterization, extension and calcification, individual ...

line 308    Re-order the three imaging modalities so they match the order in the table

line 359    The average age at diagnosis among ...

line 408    The observed discrepancies between MRI and ...

line 451     ... National Institute of Health in  Canada under grant number ...

Response 2: We sincerely thank the reviewer for their careful evaluation of our manuscript and for the insightful comments and suggestions.

  • “I was surprised that the abstract makes very little, if any, mention of the patients who were studied. Obviously this was an "enriched" population with all patients having either malignant or benign lesions.”: The reviewer is correct. This work was conducted in patients with known breast masses for the purposes of characterizing said masses only. Revision in Abstract (lines 26-29): A comparative analysis between QT3D imaging and magnetic resonance imaging (MRI) was conducted in a cohort of patients with biopsy-proven benign or malignant breast lesions comparing key metrics in quantifying breast masses for the purposes of breast mass characterization.
  • “I was surprised to discover... that the cost of a QT3D system is about $500,000... which would certainly limit its use in resource-constrained settings.”: Thank you for this important point. We recognize that while QT3D does not require contrast agents or ionizing radiation, the capital cost of the system could represent a barrier to adoption in low-resource countries or healthcare systems. At this time, we are not conducting economic analyses of this technology but limiting the work to scientific study only of image characteristics.
  • “There is extensive use of acronyms... BACT in line 170, ITK in line 191, as well as UOQ and LIQ in line 265...”: We have added a table listing all acronyms and abbreviations used throughout the manuscript.
  • “The labels in Figure 1 are essentially illegible… surely you can present a better quality image?”: We have replaced Figure 1 with a higher-resolution image that includes clearer labels.
  • “The aspect ratio of the breast cross-sections in Figure 2 seems to be distorted…”: Figure 2 has now been reformatted to maintain the correct aspect ratio.
  • “Not all automated breast ultrasound (ABUS) systems are the same... This needs to be emphasised in the manuscript.”:  We agree that this distinction is important; however, we want to emphasize that the present device being used  is a unified system that integrates both reflective and transmissive imaging. At present, it functions as a “single, combined system."
  • “No mention is made in the methods about the acquisition of mammographic data... was it 2D or 3D?”: We have updated the Methods section to include a description of mammographic data acquisition. To clarify, all mammography performed in this study was standard 2D digital mammography carried out as part of patient standard of care. Revision in Methods (Table # 3. lines 306-308): “Mammographic images were obtained using standard 2D full-field digital mammography systems as part of patient standard of care.”
  • “I was interested to see whether the authors might conclude by suggesting that QT3D could replace MRI... Is their recommendation that one approach is better than the other or are they essentially equivalent?”: We have revised the conclusions section to provide a clearer view. While our results show that QT3D performs comparably to MRI for lesion characterization in our study population, we do not recommend it as a complete substitute. The study here is limited to 19 patients only and one cannot recommend a substitution of one technology with another, which is comparable on that basis. Instead, we propose that quantitative ultrasound tomography could serve as a viable alternative or adjunct in settings where MRI is contraindicated or unavailable to the patient. Revision in Conclusion (lines 449 - 452): “Quantitative ultrasound tomography demonstrated diagnostic performance comparable to that of MRI in this study population. Tomographic ultrasound may serve as an effective alternative in scenarios where MRI is not feasible due to availability or patient contraindications.”